# Fighting Antimicrobial Resistance in Neonatal Intensive Care Units: Rational Use of Antibiotics in Neonatal Sepsis

**DOI:** 10.3390/antibiotics12030508

**Published:** 2023-03-03

**Authors:** Dimitrios Rallis, Vasileios Giapros, Anastasios Serbis, Chrysoula Kosmeri, Maria Baltogianni

**Affiliations:** 1Neonatal Intensive Care Unit, School of Medicine, University of Ioannina, 451 10 Ioannina, Greece; 2Department of Paediatrics, School of Medicine, University of Ioannina, 451 10 Ioannina, Greece

**Keywords:** early-onset sepsis, extended-spectrum beta-lactamases, late-onset sepsis, multidrug resistance, carbapenem-resistant enterobacteriaceae

## Abstract

Antibiotics are the most frequently prescribed drugs in neonatal intensive care units (NICUs) due to the severity of complications accompanying neonatal sepsis. However, antimicrobial drugs are often used inappropriately due to the difficulties in diagnosing sepsis in the neonatal population. The reckless use of antibiotics leads to the development of resistant strains, rendering multidrug-resistant pathogens a serious problem in NICUs and a global threat to public health. The aim of this narrative review is to provide a brief overview of neonatal sepsis and an update on the data regarding indications for antimicrobial therapy initiation, current guidance in the empirical antimicrobial selection and duration of therapy, and indications for early discontinuation.

## 1. Introduction

Neonatal sepsis refers to life-threatening organ dysfunction due to a dysregulated host response to infection [1]. Early onset sepsis (EOS) occurs in the first 72 h of life and is associated with intrauterine or maternal factors [2,3]. The estimated incidence of EOS in Europe was 0.28 to 2.1 episodes/1000 live births [4,5,6]. In the U.S., the incidence was 7.4/1000 in preterm, 0.76/1000 in late preterm, and 0.31/1000 in term neonates [7]. Late-onset sepsis (LOS) occurs after the first 72 h of life and is mainly associated with factors in the neonatal intensive care unit (NICU), with the greatest incidence reported between the tenth and the twenty-second day of life [8,9,10]. The incidence of LOS is inversely proportional to gestational age and birth weight. Data from the United Kingdom reported the incidence of LOS in 8 episodes/1000 live births in all neonates, in 16–30% of neonates with a very low birth weight, and in 50% of neonates with an extremely low birth weight [9,10,11].

The prompt diagnosis of neonatal sepsis is critical to prevent serious and life-threatening complications; however, it may be difficult due to nonspecific clinical manifestations and the low predictive value of infection biomarkers [12]. Uncertainty regarding the existence of a neonatal infection may result in an unnecessarily prolonged administration of antibiotics. On the other hand, the reckless use of antibiotics leads to the development of resistant strains [13], and the worldwide spread of antimicrobial resistance has emerged as a major healthcare challenge [14,15].

The aim of this narrative review is to provide a brief overview of neonatal sepsis and an update on the data regarding indications for antimicrobial therapy initiation, current guidance in the empirical antimicrobial selection and the duration of therapy, and indications for early discontinuation.

## 2. Neonatal Sepsis

### 2.1. Pathogenesis

The organisms most implicated in EOS are *Streptococcus agalactiae* (or group B *Streptococcus*, GBS) and *Escherichia coli*. These account for nearly 70% of cases in both premature and full-term neonates. Other pathogens include other streptococci (*Streptococcus viridans*, *Streptococcus pneumoniae*), *Staphylococcus aureus*, enterococci (*Enterococcus* spp.), Gram-negative enterobacteriaceae (*Enterobacter* spp.), and *Listeria monocytogenes* [9]. Fungal infections, especially *Candida* spp., are implicated in neonates with a very low birth weight and may manifest with a clinical picture of EOS, usually in the first 24 h of life [16].

The pathogens responsible for LOS include microorganisms in the hospital environment and especially in the NICU. The most implicated pathogens are coagulase-negative staphylococci (CoNS), which are responsible for 53–78% of episodes, *Staphylococcus aureus*, and Gram-negative bacteria which mainly include *Escherichia coli*, *Klebsiella* spp., *Enterobacter* spp., *Pseudomonas* spp., *Citrobacter* spp., and *Serratia* spp. The incidence of fungal infection with *Candida* spp. In neonates with a very low birth weight is significant [5,17,18].

### 2.2. Clinical Presentation

Clinical signs of neonatal infection are nonspecific and include manifestations from various systems. Findings from the respiratory system include apnea, cyanosis, grunting, nasal flare, tachypnea, and recessions of the intercostal or sublateral spaces [19,20]. Pulmonary hypertension, impaired cardiac output, poor peripheral perfusion, and hypoxemia are predominant in the cardiovascular system [21,22]. Acidosis and hypoxemia can increase the pressure of the pulmonary artery, prolonging the patency of the ductus arteriosus [21,22]. Neonates with sepsis may experience tachycardia, poor perfusion, and normal blood pressure (high systemic vascular resistance), or hypotension with adequate (warm shock, vasodilation), or insufficient perfusion (cold shock, vasoconstriction) [21,23,24]. Meningitis manifests with temperature instability, lethargy, jaundice, respiratory distress, feeding intolerance, increased irritability, a change in the level of consciousness, hypotonia, or convulsions [25,26]. In the gastrointestinal tract, abdominal distention, poor feeding, vomiting, or diarrhoea are the most common findings. Necrotizing enterocolitis, which is a multifactorial disease of mainly premature neonates, has been associated with a bacterial overgrowth of specific organisms in the immature intestinal lumen [27].

### 2.3. Biomarkers

Blood culture is the gold standard test for the diagnosis of neonatal infection [28]. The sensitivity, however, is estimated at 40% and depends on the amount of blood incubated, the use of prenatal antibiotics, the severity of bacteraemia, and the capabilities of the laboratory [29]. Significant advances have been made in molecular diagnostics for the real-time identification of pathogenic microorganisms by polymerase chain reaction (PCR) [30].

The number of white blood cells is difficult to interpret in the neonatal period because it changes significantly with the day of life and the gestational age [12,31]. A low white blood cell count, low absolute neutrophil count, and a high immature/total neutrophil ratio are associated with sepsis [28,32,33]. A leukocyte count <5000 to 7500/mm^3^ is pathological; however, although the specificity is at 91%, the sensitivity is estimated at 29%. The total number of neutrophils increases after birth and reaches maximum levels within 6–12 h of life. An immature/total leukocyte ratio >20% is indicative of infection; however, it can be affected by various non-infectious causes [28,32,33].

C-reactive protein (CRP) is one of the most available and most frequently used biomarkers [29,34,35]. CRP is an acute phase biomarker synthesized by the liver. It has a half-life of 24–48 h and needs 10–12 h to increase significantly [36]. Repeated determination of CRP 24–48 h after the onset of symptoms increases sensitivity and helps monitor the response to treatment [37,38,39,40]. Levels of CRP that remain normal in repeated measurements indicate the absence of infection with a high specificity [30,38,39,40,41]. CRP values are affected by a premature rupture of the membranes, maternal fever, meconium aspiration, and foetal distress. In premature neonates, the reference values for CRP are lower, as is its increase in response to infection [42,43].

Procalcitonin (PCT) is an acute phase protein produced by hepatocytes and macrophages. It increases 4 h after exposure to bacterial endotoxin, peaks in 6–8 h, and remains elevated for at least 24 h [12,44,45]. Its half-life is approximately 25–30 h [45]. However, even in the absence of infection, the serum concentration of PCT varies greatly as it is low at birth, peaks at 24 h, and returns to baseline at 48 h [28,32,45]. Additionally, its value increases in neonates that require resuscitation, in neonates born to mothers with chorioamnionitis, those who are colonized with GBS, and in the premature rupture of foetal membranes [30,32].

Other biomarkers, such as the neutrophil index CD64, interleukin IL-6, and IL-8, have only been applied in research as they increase very quickly after a bacterial infection but return to normal levels before 24 h, limiting their clinical use [12,29,30,44].

### 2.4. Diagnosis

The diagnosis of both EOS and LOS involves difficulty as sepsis in neonates usually manifests atypically [46,47,48]. Multiple common, non-infectious conditions have a similar clinical presentation to that of neonatal infection, while laboratory tests are of limited value due to their low sensitivity and the variability of normal limits in the neonatal period [49]. Therefore, a combination of clinical signs and laboratory findings is necessary for an optimal diagnosis. According to an international consensus for the diagnosis of neonatal infection, the existence of a positive blood culture or at least laboratory and clinical criteria is a prerequisite [50,51,52,53]. Therefore, episodes are classified, according to the certainty of diagnosis, into (a) confirmed sepsis, where there is a clinical suspicion (clinical signs) with a positive blood culture or PCR, (b) probable sepsis, when the blood culture is negative but there are pathological clinical and laboratory findings, and (c) suspected sepsis, where there is only the presence of risk factors but a negative blood culture and absent clinical signs [50,51,52].

### 2.5. Prognosis

#### 2.5.1. Mortality

Mortality from neonatal sepsis depends on the degree of prematurity and individual morbidity. In neonates with a very low birth weight, up to 20% of mortality is associated with sepsis [3]. While mortality rates are significantly lower in full-term neonates, sepsis still plays an important role in this group, especially if conditions such as immunodeficiency, meconium aspiration, galactosemia, and underlying cardiac or pulmonary abnormalities coexist [3].

#### 2.5.2. Long-Term Neurological Complications

Multiple studies have associated neonatal infection with brain damage, neurodevelopmental delay, and cerebral palsy [54,55,56,57]. Early infection doubles the risk of complications, such as the occurrence of bronchopulmonary dysplasia, periventricular leukomalacia, intra-ventricular haemorrhage, and the retinopathy of prematurity [54,55,56]. Chorioamnionitis and the increased risk of cerebral palsy are not only limited to premature neonates but also full-term neonates. The most common lesion is white matter damage, which is characterized by focal cystic periventricular leukomalacia and/or diffuse necrosis, while any form of neonatal infection, including clinical infection, meningitis with or without sepsis and necrotizing enterocolitis with or without sepsis, is associated with an increased risk of neurodevelopmental delay [54,56].

## 3. Approaching Neonates Suspected of Sepsis to Avoid Misuse of Antibiotics

Based on the gestational age of the neonate, the risk assessment for EOS and management can be performed with the following different approaches [58,59,60].

### 3.1. Approaching Neonates Suspected of EOS >35 Weeks of Gestational Age

#### 3.1.1. Assessment of Risk Factors

Neonates can be categorized according to their risk for EOS based on perinatal risk factors and the clinical assessment of the neonate [58,59]. This approach results in a higher number of low-risk neonates receiving antibiotics compared to the other two approaches. This approach is based on the assessment of red or non-red flag risk factors and/or clinical indications, and neonates are either (a) screened and started on antimicrobial treatment, (b) closely monitored, or (c) receive routine neonatal care.

#### 3.1.2. Multifactorial Risk Assessment with Clinical and Laboratory Scores

In this case, the estimate is based on the calculation of a risk score (Kaiser Sepsis Calculator, https://neonatalsepsiscalculator.kaiserpermanente.org (accessed on 10 January 2023) assessing specific parameters for neonates >35 weeks’ gestation [58,61,62,63]. The duration of gestation, the GBS colonization of the mother, the higher temperature of the mother during childbirth, the duration of rupture of the foetal membranes, and the type and duration of antibiotic treatment during childbirth are introduced into the model. The recommendation for screening and treatment, close monitoring, or routine care is based on the estimated risk of early infection calculated according to the regional incidence of EOS. The American Academy of Pediatrics endorsed the Kaiser Sepsis Calculator for the management of EOS in neonates ≥34 weeks’ gestation [60], and several institutions have reported a significant reduction in antibiotic administration after adopting that approach [63,64,65,66]. However, further research regarding safety is warranted to take into account the differences in practices between healthcare services.

#### 3.1.3. Risk Assessment Based on the Clinical Picture

According to the third approach, perinatal factors are not considered. In the case of suspected infection, the neonate (>35 weeks of gestation) is evaluated based on the clinical presentation during the first 48 h of life, and the decision for laboratory screening and the initiation of antimicrobial treatment is determined accordingly [20,58,67]. Asymptomatic, full-term neonates with an abnormal blood count or CRP should not be transferred to the NICU and treated unless there is another reason because the positive prognostic value of these laboratory tests is too low. Those neonates should be closely monitored and rescreened at 6–12 h. In addition, asymptomatic, full-term neonates are extremely unlikely to have sepsis, regardless of risk factors. It should be noted that if this approach is chosen, the clinical assessment of the neonate should be repeated regularly and at relatively short intervals in a standardized manner. Repeated clinical assessments should be recorded in detail and the criteria for initiating antimicrobial treatment defined in advance [58].

### 3.2. Approaching Neonates Suspected of EOS ≤34 6/7 Weeks of Gestational Age

Neonates of gestational age ≤34 6/7 weeks can be categorized in terms of risk, depending on the conditions of preterm birth, and treated accordingly [60]. This approach aims to avoid the use of antibiotics in a group of premature neonates who are at very low risk for EOS.

Neonates at high risk for EOS are those born prematurely due to cervical insufficiency, premature onset of childbirth, premature rupture of foetal membranes, intrauterine infection, and/or sudden and unexplained foetal distress. They should have a blood culture performed and should be administered empirical antimicrobial treatment.

Premature neonates at a relatively low risk for early sepsis are those born under the following conditions and who meet all the following criteria: (a) were born prematurely due to maternal and/or foetal signs (such as preeclampsia or non-infectious maternal disease, placental insufficiency, or foetal growth restriction), (b) were born by Caesarean section, and (c) those without prior onset or induction of birth or rupture of the membranes before birth. These neonates can initially be treated either without laboratory screening and empirical administration of antimicrobial therapy or with blood culture and close clinical observation. For neonates without improvement after initial stabilization and/or severe systemic instability, empirical administration of antibiotics is a reasonable but not mandatory alternative.

In premature neonates born due to maternal and/or foetal signs of vaginal delivery or Caesarean section following the induction of labour and/or the rupture of the foetal membranes prior to delivery, factors related to the pathogenesis of GBS disease should be considered. If the mother has an indication for GBS chemoprophylaxis and has not received appropriate chemoprophylaxis (penicillin, ampicillin, or cefazoline ≥4 h before delivery), or if an infection is suspected during delivery, the neonate should be treated as a high-risk premature neonate. Otherwise, an acceptable approach is: (a) close monitoring for neonates in good general condition at birth, and (b) blood culture and initiating antimicrobial treatment in neonates with postpartum respiratory and/or cardiovascular instability.

### 3.3. Approaching Neonates Suspected of LOS

The approach is based on the assessment of risk factors and the existence of clinical signs, as previously mentioned [19,68]. Laboratory screening that includes a complete blood count, CRP, urine culture, and a lumbar puncture based on indications is recommended, while empirical antibiotic treatment should be initiated [19,68].

## 4. Antimicrobial Therapy

### 4.1. Choice of Antimicrobial Agent

The early administration of appropriate antibiotics is the cornerstone in the treatment of neonatal sepsis. The appropriate empirical selection of antibiotics for EOS is based on the possible pathogens [58,60,69]. The combination of ampicillin and gentamicin is the most appropriate treatment for the most common organisms, GBS and *Escherichia coli* [58,60,69]. If meningitis is suspected, an expanded-spectrum cephalosporin such as cefotaxime should substitute the aminoglycoside [70,71]. Ceftriaxone should not be used in neonates because it increases serum bilirubin [72].

The recommended first-line treatment for LOS is flucloxacillin (or ampicillin) in combination with gentamicin [58,60,69]. The second-line treatment is vancomycin or teicoplanin, along with another antibiotic with a broad-spectrum efficacy against Gram-negative bacteria, such as piperacillin/tazobactam. When a specific pathogen is isolated, treatment should be targeted (Table 1) [58,60,69,70,71].

### 4.2. Risk of Antimicrobial Therapy

Microbiome changes from antibiotics administered in early life are associated with chronic diseases in later life. The prolonged antibiotic exposure of premature neonates is associated with an increased risk of LOS, increased morbidity and mortality, and an increased risk of chronic lung disease, retinopathy of prematurity, periventricular leukomalacia, and necrotizing enterocolitis [13,65,73,74,75,76]. In addition, admission to the NICU for laboratory assessment and the empirical administration of antibiotics to neonates with risk factors entails the separation of the neonate from their mother, delaying the development of a maternal–neonate bond and the establishment of breastfeeding [13,65,74,75,76].

Nearly half of the pathogens causing neonatal sepsis exhibit a high resistance to first-line antibiotics such as ampicillin, gentamicin, and cefotaxime [77]. Moreover, Gram-negative bacteria show a multidrug-resistant (MDR) phenotype, suggesting the greatest concern in the neonatal population [77]. Neonates with prolonged hospitalization are at high risk of infection due to MDR pathogens [78]. The recent global report on the epidemiology and burden of sepsis, as well as the reports by the BARNARDS and the GARDP studies, highlight the increasing attribution of antimicrobial resistance to neonatal sepsis mortality [79,80].

Previously, MDR Gram-positive bacteria were predominant in NICUs. However, over the last decades, the isolation of MDR Gram-negative bacteria has increased [78,81,82]. Of note, the resistance of Gram-negative bacteria to traditional antibiotics is increasing [83]. This is due in part to the plasmid-mediated intergenic transfer of new resistance genes that has been observed among resistance genes for extended-spectrum beta-lactamases (ESBL) between *Escherichia coli* and *Klebsiella pneumoniae*, as well as among other enterobacteriaceae [84]. MDR Gram-negative bacteria are the greatest concern in the neonatal population and have very limited therapeutic options [84,85]. According to the World Health Organization (WHO), the most common emerging MDR bacteria include carbapenem-resistant *Acinetobacter baumannii* and *Pseudomonas aeruginosa*, and carbapenem-resistant and third-generation cephalosporin-resistant enterobacteriaceae [86].

The inappropriate and excessive use of broad-spectrum antibiotics has been associated with the emergence of antibiotic-resistant pathogens. Additionally, enterobacteriaceae pathogens producing ESBL and carbapenem-resistant enterobacteriaceae (CRE) are implicated in outbreaks of infection in NICUs and are associated with increased morbidity and mortality [78]. The antibiotics most commonly leading to resistance include third-generation cephalosporins (e.g., cefotaxime, ceftazidime, and ceftriaxone), extended-spectrum penicillins (e.g., ticarcillin/clavulanate, piperacillin/tazobactam), carbapenems (e.g., meropenem and imipenem) and quinolones (e.g., ciprofloxacin). Third-generation cephalosporins have also been associated with an increased risk of fungal sepsis in neonates with a very low birth weight and should be avoided when meningitis has been excluded [87].

In that aspect, antibiotic therapy should be performed with narrow-spectrum antibiotics if possible and only used with a strong suspicion of infection. In a recent study in the United Kingdom, the majority of the pathogens responsible for EOS were susceptible to the combination of penicillin and gentamicin [87]. Except for CoNS, the majority of the pathogens implicated in LOS (*Staphylococcus aureus*, *Escherichia coli*, *Enterococcus* spp., enterobacteriaceae) were susceptible to flucloxacillin and gentamicin. More recent surveillance data has demonstrated that the combination of flucloxacillin and gentamicin has become more effective against LOS pathogens than amoxicillin and cefotaxime due to the relatively high resistance rate of enterobacteriaceae to cephalosporins [87].

### 4.3. Common Antibiotic-Resistant Pathogens in Neonatal Sepsis

#### 4.3.1. Extended-Spectrum Beta-Lactamase-Producing Enterobacteriaceae

Resistance to beta-lactam antibiotics was first found in *Klebsiella pneumoniae* [88], and this resistance was shown to derive from the production of ESBL enzymes able to hydrolyse the beta-lactam ring [89]. In recent decades, the prevalence of ESBL-producing *Klebsiella pneumoniae* has greatly increased. Factors associated with ESBL sepsis include prematurity, low birth weight, prolonged neonatal hospitalization, and prolonged antibiotic administration [89]. Furthermore, resistance to other antibiotic options, including quinolones, can lead to treatment difficulties [90], whereas the resistance of enterobacteriaceae to third-generation cephalosporins has been estimated up to 39.5% in neonates and in infants up to 12 months of age, according to European data [91].

#### 4.3.2. Carbapenem-Resistant Enterobacteriaceae

Carbapenems were used for infections caused by ESBL-producing pathogens, with carbapenem-resistant *Klebsiella pneumoniae* first emerging during the 1990s [92]. The worldwide expansion of CREs has been increasingly reported [93,94]. However, limited data has been published regarding CRE infections in neonates. Outbreaks have been reported in NICUs, and the geographic distribution is broadly comparable with the data reported in adults [95].

#### 4.3.3. Colistin-Resistant Enterobacteriaceae

Given that the prevalence of CRE has been increased worldwide, polymyxins have become an important alternative [96]. With the overuse of colistin for the treatment of MDR Gram-negative bacterial infections, however, the presence of plasmid-mediated colistin-resistant *Acinetobacter* spp., *Pseudomonas* spp., and enterobacteriaceae have been reported worldwide [97,98]. Colistin resistance in *Klebsiella pneumoniae* has been reported from numerous regions, including Europe, North America, South America, Asia, and South Africa [97,99,100]. Evidence suggests that inappropriate use of colistin, such as suboptimal dosing or prolonged monotherapy, may contribute to the emergence of colistin resistance, even with pathogens initially susceptible to colistin [101].

### 4.4. Novel Therapies for Resistant Pathogens

Glycopeptides remain an appropriate treatment for most staphylococcal infections [102]. In the neonatal population, CoNS and *Staphylococcus aureus* strains are susceptible to vancomycin, whereas an increase in vancomycin minimum inhibitory concentration (MIC) values has been reported for the methicillin-resistant *Staphylococcus aureus* within the susceptible range. In Gram-positive infections unresponsive to treatment, linezolid has been the best alternative, with daptomycin as another option for persistent staphylococcal sepsis [103].

Among newer promising antibiotics, ceftaroline and ceftobiprole are cephalosporins active against MDR staphylococci [104], whereas oritavancin, dalbavancin, and telavancin are lipoglycopeptides active against MDR Gram-positive pathogens [105]. Ceftarolin was evaluated in a Phase 2 study in neonates with LOS and was found to have a pharmacokinetics and safety profile comparable to previous paediatric data [106]. Ceftobiprole has been evaluated in a Phase 1 study in neonates. The pharmacokinetic parameters were similar to those of adults, and the drug was well-tolerated [107]. Among available lipoglycopeptides, dalbavancin’s pharmacokinetics and pharmacodynamics were examined for children younger than 6 years old, and a single dose was found to have adequate pharmacodynamic target attainment [108].

In Gram-negative sepsis, many ESBL-producing pathogens may produce multiple beta-lactamases simultaneously, and thus reduce the potential efficacy of the beta-lactamase inhibitors [78]. For Gram-negative infections with severely antibiotic-resistant pathogens, carbapenems have become the cornerstone of treatment. Meropenem is the most commonly used agent, and doripenem is a newer carbapenem with efficacy against *Pseudomonas aeruginosa* [109]. Carbapenems should be used as monotherapy since there is no evidence for increased efficacy when combined with aminoglycosides [110].

The increased resistance to carbapenems has led to the use of traditional agents, such as colistin, fosfomycin, and tigecycline [102,110]. Colistin use has been mostly reported in neonatal sepsis; however, evidence is still limited with respect to its use in specific infections caused by CRE [102,110]. Of note, *Proteus* spp. and *Serratia* spp. have been reported to be intrinsically resistant to colistin [111]. In addition, fosfomycin is less experienced in neonatal sepsis, comprising a final option for MDR Gram-negative bacteria [102,110]. Fosfomycin is active against the majority of CRE pathogens, while it achieves excellent concentrations in body fluids. When fosfomycin is used as monotherapy, resistance can develop rapidly and it should therefore be combined with other agents [102,110]. Tigecycline demonstrates good efficacy against MDR Gram-positive and Gram-negative bacteria, but not against *Pseudomonas aeruginosa*. However, its use in neonatal sepsis is limited, given the possible side effects on bone growth [112,113]. Finally, newer antibiotics specifically targeting infections caused by CRE include aztreonam/avibactam, vaborbactam/meropenem (carbavance), and plazomicin; however, they have limited evidence in neonates (Table 2) [102,106,107,108].

## 5. Antimicrobial Stewardship in Neonatal Units

Antimicrobial stewardship refers to those measures focused on monitoring and controlling optimal antimicrobial use. Through coordinated interventions, the appropriate use of antimicrobials is promoted and measured by selecting antimicrobials that are most specific for isolated pathogens with a narrow spectrum, in the optimal doses, and for the optimal duration [87,114]. In addition, measures promoting infection prevention and control, including hand hygiene, limitations of the visitors, vaccination of personnel, interventions related to infrastructure, and isolation measures when required, lead to a reduction in the incidence of healthcare-associated infections and thus a lower antibiotic use [115]. Additionally, evidence suggests that when there is poor compliance with hand hygiene, overcrowding, and inadequate cleaning of equipment infections are more common, especially due to Gram-negative pathogens. Nosocomial infections are also more frequent when there is a delay in the introduction of breast milk and when a longer time is taken to establish full enteral feeds.

NICUs should aim to develop effective antimicrobial stewardship, including (a) establishing coordination with other teams such as microbiology, infection control, paediatric infectious diseases, and pharmacists, (b) establishing a surveillance system of bloodstream infections, (c) developing guidance for optimal use of narrow spectrum empiric antibiotic whenever possible, and (d) regular audits of compliance [87].

### 5.1. Principles of Antimicrobial Stewardship

Antibiotic susceptibility testing should determine, with the following steps, whether antibiotics should be continued, modified, or discontinued. Colonization should be distinguished from true infection and not be treated [116,117].

Susceptibility results should guide clinicians to treat with narrow-spectrum antibiotics that are more effective with fewer side effects. For instance, oxacillin instead of vancomycin should be the agent of choice for neonatal sepsis due to methicillin-susceptible Staphylococcus aureus. Additionally, when resistant pathogens are detected, the empiric antibiotics should be accordingly modified (i.e., when a Gram-negative pathogen is detected, vancomycin should be stopped) [116,117].

Clinicians should also modify antibiotic treatment based on the MIC, especially when antibiotics are known to have a reduced penetration in specific sites. Antibiotic agents with MICs far from the clinical breakpoint should be used as adequate tissue levels are warranted [116,117].

### 5.2. Duration of Antimicrobial Therapy

When treatment has been administered due to risk factors but the neonate remains asymptomatic and the blood cultures are negative at 36–48 h, it is reasonable to discontinue treatment. It is very unlikely that a blood culture that becomes positive after more than 48 h in an asymptomatic neonate is of clinical significance [13,73,76]. If there was a clinical impression of possible sepsis at the initiation of treatment and the blood cultures are negative, a longer duration of treatment may be justified, usually for five days [73,76]. If blood cultures are positive, treatment should last for at least 10 days, while in the case of meningitis, treatment may be required for at least 21 days (Table 1) [65]. Osteomyelitis, endocarditis, and deep abscesses that cannot be surgically excluded may require several weeks of treatment [65].

However, there is no definitive evidence regarding the optimal duration of antibiotic treatment for culture-proven neonatal sepsis. Evidence reports that the duration of treatment may vary from a minimum duration of 7 days to a maximum of 14 days for culture-proven, uncomplicated neonatal septicemia [13]. A shorter duration of antibiotic administration has been examined in neonates with culture-positive sepsis in previous randomized trials [118,119,120] and is currently being examined in an ongoing clinical trial [121]. Chowdhury et al. reported a treatment failure within 28 days of observation in five neonates in the 7 day treatment group compared to one in the 14 day treatment group [120]. Similarly, when a 7 day course of antibiotics was examined in comparison to ten days among culture-positive sepsis neonates, Rohatgi et al. reported a similar treatment failure between groups [119]. In a comparison of 10 days versus 14 days of therapy among culture-positive neonates, Gathwala et al. concluded that the outcome was similar in both study groups, with one treatment failure in each group [118]. Additionally, an ongoing trial by Dutta et al. has been designed to examine whether a total of 7 days of antibiotics is not inferior compared to a total of 14 days of antibiotics among neonates with uncomplicated, culture-proven sepsis with respect to definite or probable relapse of sepsis within 21 days of observation after intravenous antibiotic completion [121]. Finally, Keij et al. examined the efficacy and safety of early intravenous-to-oral antibiotic switch therapy compared with a 7 day course of intravenous antibiotics among neonates with probable bacterial infection [122]. After an initial 48–72 h of intravenous antibiotics, neonates were switched to an oral suspension of amoxicillin/clavulanic acid (intervention group) or continued on intravenous antibiotics, according to the local protocol, for a total of 7 days. Neonates were included in the trial if they were of a postmenstrual age of 35 weeks or older and a postnatal age of 0–28 days with a body weight of at least 2 kg, whereas neonates with culture-proven bacterial sepsis or severe clinical sepsis were excluded. The authors reported that a cumulative reinfection rate at 28 days did not differ between the groups (<1%) [122]. Of note, the inclusion of neonates with specific gestational age and birthweight in the previous trials suggested that the studies applied to a selected group of neonates. Therefore, further evidence is warranted before generalizing. Each case should be carefully reviewed by the clinicians, and the duration of therapy should be individually fine-tuned.

## 6. Conclusions

Neonatal sepsis is a leading cause of morbidity and mortality in neonates. It is associated with white matter damage and impaired neurodevelopmental outcome. The management of neonatal sepsis is based on risk factors and clinical signs. In the case of exclusion of infection, empirical antimicrobial treatment can be discontinued at 36–48 h as prolonged administration of antimicrobial treatment in the absence of infection is associated with a worse prognosis. The current guidelines for antibiotic use in neonatal sepsis are summarized in Table 3. Judicious use of antibiotic therapy, with narrow-spectrum antibiotics and when only used with a strong suspicion of infection, is essential for fighting multi-resistant pathogens. NICUs should develop antibiotic stewardship programs to ensure optimal antibiotic use.

## Figures and Tables

**Table 1 antibiotics-12-00508-t001:** Targeted antimicrobial therapy on neonatal sepsis.

Pathogen	Agent	Duration of Treatment
Group B *Streptococcus*	Ampicillin and gentamicin	Bacteremia 10 days Meningitis 14 days Septic arthritis/osteomyelitis 3–4 weeks Endocarditis at least 4 weeks
*Escherichia coli*, other gram-negative bacilli (*Citrobacter* spp., *Enterobacter* spp. and *Serratia* spp.)	Ampicillin and gentamicin or Cefotaxime and gentamicin	Bacteremia 14 days Meningitis 21 days
*Listeria monocytogenes*	Ampicillin and gentamicin	Bacteremia 10–14 days Meningitis 14–21 days
*Coagulase negative Staphylococcus*	Vancomycin or teicoplanin	Bacteremia 10 days
*Staphylococcus aureus*	Nafcillin or oxacillin	Bacteremia 10 days

**Table 2 antibiotics-12-00508-t002:** Novel therapies for resistant pathogens in neonatal sepsis.

Agent	Pathogen	Notes
Vancomycin	CoNS/oxacillin-resistant *Staphylococcus aureus*	
Teicoplanin	CoNS/oxacillin-resistant *Staphylococcus aureus*	
Linezolid	CoNS/oxacillin-resistant *Staphylococcus aureus*	Unresponsive Gram infections
Daptomycin	CoNS/oxacillin-resistant *Staphylococcus aureus*	Persistent staphylococcal bacteremia
Ceftaroline	MDR staphylococci	
Ceftobiprole	MDR staphylococci	
Oritavancin/dalbavancin/telavancin	MDR Gram-positive bacteria	
Piperacillin/tazobactam	Most ESBL enterobacteriaceae	Poor cerebrospinal fluid penetration
Meropenem	ESBL enterobacteriaceae	Monotherapy recommended
Doripenem	ESBL enterobacteriaceae	Great activity against *Pseudomonas aeruginosa*
Ciprofloxacin	ESBL enterobacteriaceae	
Colistin	CRE	
Fosfomycin	CRE, EDR gram negative bacteria	Should not be used as monotherapy
Tigecycline	CRE, MDR Gram-positive, MDR Gram-negative	Inactive against *Pseudomonas aeruginosa*
Aztreonam/avibactam	CRE	Limited data in neonates
Carbavance (vaborbactam/meropenem)	CRE	Limited data in neonates
Plazomicin	CRE	Limited data in neonates

CoNS—Coagulase-negative Staphylococcus; MDR—multidrug-resistant; ESBL—extended-spectrum beta-lactamase; EDR—extremely drug-resistant; CRE—Carbapenem-resistant Enterobacteriaceae.

**Table 3 antibiotics-12-00508-t003:** Guidelines for antibiotic use in neonatal sepsis.

Empirical therapy for EOS is ampicillin and gentamicin; Substitute cefotaxime for aminoglycoside in suspected meningitis; Empirical therapy for LOS is flucloxacillin or ampicillin plus gentamicin; Second-line treatment for LOS is vancomycin or teicoplanin plus piperacillin/tazobactam; Glycopeptides should be used for staphylococcal infections; methicillin-susceptible *Staphylococcus aureus* should be treated with oxacillin; Resistant staphylococcal infections can be treated with linezolid or daptomycin; Novel therapies for MDR staphylococci are cephalosporins such as ceftaroline and ceftobiprole and lipoglycopeptides such as oritavancin, dalbavancin, and telavancin; ESBL-producing pathogens can be treated with carbapenems (meropenem, doripenem); After isolation of specific pathogens, treatment should be changed to the narrowest spectrum, in the optimal doses, and for the optimal duration.

EOS—early onset sepsis; LOS—late onset sepsis; MDR—multidrug-resistant; ESBL—extended-spectrum beta-lactamase.

## Data Availability

Not applicable.

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
