# Peer review of "Fighting Antimicrobial Resistance in Neonatal Intensive Care Units: Rational Use of Antibiotics in Neonatal Sepsis"

_antibiotics, 2023, doi:10.3390/antibiotics12030508_

Round 1

Reviewer 1 Report

Kindly consider addressing the following comments and suggestions:

1.      I would suggest modifying the title to “Fighting antimicrobial resistance in neonatal intensive care units: rational use of antibiotics in neonatal sepsis.”

2.      The aim of this review was to provide an update on the guidelines for antimicrobial therapy in neonates. However, most of the cited references are outdated. Some information is not supported by a reference, and others are supported by the wrong citation. Examples are given below:

·         Lines 29-34: Recent epidemiological study reports on early-onset sepsis are available in the literature for countries such as the United States, the United Kingdom, Norway, Sweden, and others.

·         Lines 70-87: Most of the supporting references are outdated. In addition, the information in lines 82-86 requires a reference citation.

·         Lines 122-134: Information on the diagnosis of neonatal sepsis can be supported by more recent reports, such as “WHO Sepsis Technical Expert Meeting 2018” and the like.

·         Line 235-237: The cited reference is from 2016. Recent reports on 2021 and 2022 are available.

·         Lines 244-245: Currently, ….. The cited reference is from 2015.

·         Lines 270 and 276: Reference [85] reports the colistin resistance in Klebsiella pneumoniae. Please cite the correct reference.

·         Lines 308-311: These agents are……… in ongoing clinical trials. The cited reference is from 2015. What is the update on these clinical trials after 8 years? The same reference was used for generating Table 2 despite the availability of more recent reports.

·         …… etc.

The authors are therefore encouraged to update and correct the cited references throughout the manuscript.

3.      The manuscript contains errors in section numbering, e.g., section 3.3 is missing and section 4.6 is used twice. I would also suggest restructuring the manuscript for a better presentation. The first three paragraphs of the introduction can be combined into one. Sections 4.3, 4.4, and 4.5 can be categorized under the heading “common antibiotic-resistant pathogens in neonatal sepsis.” Antimicrobial stewardship can be presented in a separate heading. Prognosis can be included in subsection 2.5 in section 2. Neonatal sepsis.

4.      The authors are encouraged to provide a summary of the updated guidelines for antibiotic use in NICUs, may be in a table or a figure, to highlight the main messages of the review.  

5.      Throughout the manuscript, please use the correct format of microbial nomenclature.

6.      Lines 45-47: Consider rephrasing because the use of “However” and “whereas” is compromising the meaning.

7.      Line 225: “…increased mortality and mortality, …” I think the authors mean morbidity and mortality. Please correct.

Author Response

Answers to the reviewers' comments:

We would like to thank the editor and all the reviewers for their constructive comments on our paper. We have outlined our responses to the various points raised by the reviewers sequentially and have indicated changes made to the manuscript as lines. In the revised manuscript all the changes are highlighted in red.

Reviewer 1

Kindly consider addressing the following comments and suggestions: 

  1. I would suggest modifying the title to “Fighting antimicrobial resistance in neonatal intensive care units: rational use of antibiotics in neonatal sepsis.”

Reply:

We thank the reviewer for raising this point.

The title has been accordingly modified to “Fighting antimicrobial resistance in neonatal intensive care units: rational use of antibiotics in neonatal sepsis.”

Please see lines 2-3.

  1. The aim of this review was to provide an update on the guidelines for antimicrobial therapy in neonates. However, most of the cited references are outdated. Some information is not supported by a reference, and others are supported by the wrong citation. Examples are given below:

Reply:

We thank the reviewer for raising this point.

The cited references have been replaced with more recent references as shown below.

  • Lines 29-34: Recent epidemiological study reports on early-onset sepsis are available in the literature for countries such as the United States, the United Kingdom, Norway, Sweden, and others.

Reply:

We thank the reviewer for this point.

This paragraph has been updated with more recent epidemiolocal data on the incidence of EOS. Specifically: The estimated incidence of EOS in Europe was 0.28 to 2.1 episodes / 1000 live births (Giannoni 2018, Juliana 2022, Sikias 2022). In USA the incidence was 7.4 / 1000 in preterm, 0.76 / 1000 in late preterm, and 0.31 per 1000 in term neonates (Joshi 2022).

As per reviewer’s recommendation, more recent references have been cited:

(Giannoni, Agyeman et al. 2018, Joshi, Huynh et al. 2022, Juliana, Holband et al. 2022, Milton, Gillespie et al. 2022, Sikias, Biran et al. 2022).

Please see lines 39-52.

  • Lines 70-87: Most of the supporting references are outdated. In addition, the information in lines 82-86 requires a reference citation.

Reply:

We thank the reviewer for this point.

As per reviewer’s recommendation, references have been updated.

Please see lines 90-105.

  • Lines 122-134: Information on the diagnosis of neonatal sepsis can be supported by more recent reports, such as “WHO Sepsis Technical Expert Meeting 2018” and the like.

Reply:

We thank the reviewer for this point.

The information on the diagnosis of neonatal sepsis has been supported by more recent reports, specifically: (Iroh Tam and Bendel 2017, Brown, Meader et al. 2020, Celik, Hanna et al. 2022, Flannery and Puopolo 2022).

Please see lines 143-155.

  • Line 235-237: The cited reference is from 2016. Recent reports on 2021 and 2022 are available. 

Reply:

We thank the reviewer for this point.

As per reviewer’s recommendation, references have been updated.

Please see lines 286-291.

  • Lines 244-245: Currently,….. The cited reference is from 2015.

Reply:

We thank the reviewer for this point.

As per reviewer’s recommendation, references have been updated.

Please see lines 298-299.

  • Lines 270 and 276: Reference [85] reports the colistin resistance in Klebsiella pneumoniae. Please cite the correct reference.

Reply:

We thank the reviewer for this point.

As per reviewer’s recommendation, references have been updated. The correct references have been used in the revise text. (Ferreira, da Silva et al. 2018, Zhan, Xu et al. 2021).

Please see lines 326-331.

  • Lines 308-311: These agents are……… in ongoing clinical trials. The cited reference is from 2015. What is the update on these clinical trials after 8 years? The same reference was used for generating Table 2 despite the availability of more recent reports.

Reply:

We thank the reviewer for raising this important this point.

Ceftarolin was evaluated in a phase 2 study in neonates with LOS and found to have pharmacokinetics and safety profile comparable to previous pediatric data (Bradley 2020). Ceftobiprole has been evaluated in a phase 1 study in neonates and the pharmacokinetic parameters were similar to those of adults, while the drug was well tolerated (Rubino 2021). Among available lipoglycopeptides, dalbavancin’s pharmacokinetics and phar-macodynamics were examined for children younger than 6 years old and a single dose found to have adequate pharmacodynamic target attainment (Carrothers 2023).

Please see lines 366-375.

  • …… etc.

The authors are therefore encouraged to update and correct the cited references throughout the manuscript.

Reply:

We thank the reviewer for this point.

As per reviewer’s recommendation, references have been replaced with more recent ones. Specifically:

  • Ref: (Pariente 2022, Talaat, Zayed et al. 2022)
  • Ref: (Flannery, Edwards et al. 2022, Juliana, Holband et al. 2022, Mariani, Parodi et al. 2022)
  • Ref: (van Duin, Arias et al. 2020, Jing, Yan et al. 2022)
  • Ref (Ding, Wang et al. 2019)
  • Ref: (El-Sayed Ahmed, Zhong et al. 2020, Liu, Wu et al. 2022)
  • Ref: (Aris, Robatjazi et al. 2020, Liu, Wu et al. 2022, Narimisa, Goodarzi et al. 2022)
  • Ref: (Rose, Fantl et al. 2021)
  • Ref: (Gupta, Yu et al. 2022)

  1. The manuscript contains errors in section numbering, e.g., section 3.3 is missing, and section 4.6 is used twice. I would also suggest restructuring the manuscript for a better presentation. The first three paragraphs of the introduction can be combined into one. Sections 4.3, 4.4, and 4.5 can be categorized under the heading “common antibiotic-resistant pathogens in neonatal sepsis.” Antimicrobial stewardship can be presented in a separate heading. Prognosis can be included in subsection 2.5 in section 2. Neonatal sepsis.

Reply:

We thank the reviewer for this point.

The headings of the manuscript have been corrected accordingly.

The first three paragraphs of the introduction were combined into one, and the manuscript has been restricted as per reviewer’s recommendation.

  1. The authors are encouraged to provide a summary of the updated guidelines for antibiotic use in NICUs, may be in a table or a figure, to highlight the main messages of the review.  

Reply:

We thank the reviewer for this point. We structured Table 3 depicting the main messages of the review.  

·      Empirical therapy for EOS is ampicillin and gentamicin

·      Add cefotaxime in suspected meningitis

·      Empirical therapy for LOS is flucloxacillin or ampicillin plus gentamicin

·      Second-line treatment for LOS is vancomycin or teicoplanin plus piperacillin/tazobactam

·      Glycopeptides should be used for staphylococcal infections, methicillin-susceptible Staphylococcus aureus should be treated with oxacillin

·      Resistant staphylococcal infections can be treated with linezolid or daptomycin

·      Novel therapies for MDR staphylococci are cephalosporins such as ceftaroline and ceftobiprole and lipoglycopeptides such as oritavancin, dalbavancin and telavancin.

·      ESBL-producing pathogens can be treated with carbapenems (meropenem, doripenem)

·      After isolation of specific pathogens, treatment should be changed to the narrowest spectrum, in the optimal doses and for the optimal duration.

  1. Throughout the manuscript, please use the correct format of microbial nomenclature.

Reply:

We thank the reviewer for this point.

As per reviewer’s recommendation, we have revised the format of microbial nomenclature throughout the manuscript.  

  1. Lines 45-47: Consider rephrasing because the use of “However” and “whereas” is compromising the meaning. 

Reply:

We thank the reviewer for this point.

This has been rephrased in the revised text.

Please see line 64.

  1. Line 225: “…increased mortality and mortality, …” I think the authors mean morbidity and mortality. Please correct.

Reply:

We thank the reviewer for this point.

This has been corrected in the revised text.

Please see line 275.

Reviewer 2 Report

This paper is well written and provides an important contribution to neonatal sepsis and antimicrobial therapy.

Author Response

Reviewer 2

This paper is well written and provides an important contribution to neonatal sepsis and antimicrobial therapy.

Reply:

We thank the reviewer.

Reviewer 3 Report

Thanks for this manuscript, I enjoyed reading that. I must admit that I missed a bit of flow in the abstract and beginning of the introduction (which should be condensed- I recommend a proofread by a native speaker). That is a pity, as it discourage interested readers of the remainder of the text (which is super interesting).

Regarding antimicrobial therapy:

1) Could you outline what the regimen for meningitis should be, is that really amp/gent + cefotaxime as you suggest in lines 212-14? Furthermore, what is your vision on replacing cefotax by ceftriax?

2) Could you elaborate on PK data pertaining the neonatal population regarding the agents you outline in table 2?

Author Response

Reviewer 3

Thanks for this manuscript, I enjoyed reading that. I must admit that I missed a bit of flow in the abstract and beginning of the introduction (which should be condensed- I recommend a proofread by a native speaker). That is a pity, as it discourages interested readers of the remainder of the text (which is super interesting).

Reply:

We thank the reviewer for this point.

As per reviewer’s recommendation, abstract and introduction have been revised.

Please see lines 15-63.

Regarding antimicrobial therapy:

  • Could you outline what the regimen for meningitis should be, is that really amp/gent + cefotaxime as you suggest in lines 212-14? Furthermore, what is your vision on replacing cefotax by ceftriax?

Reply:

We thank the reviewer for this point.

The appropriate regimen for meningitis is ampicillin, aminoglycoside and cefotaxime. Ceftriaxone should not be used in neonates, because it increases serum bilirubin (Donnelly, Sutich et al. 2017).

Please see lines 261-262.

2) Could you elaborate on PK data pertaining the neonatal population regarding the agents you outline in table 2?

Reply:

We thank the reviewer for this point.

PK data have been evaluated by recent trials as previously reported. Ceftarolin was evaluated in a phase 2 study in neonates with LOS and found to have pharmacokinetics and safety profile comparable to previous pediatric data (Bradley 2020). Ceftobiprole has been evaluated in a phase 1 study in neonates and the pharmacokinetic parameters were similar to those of adults, while the drug was well tolerated (Rubino 2021). Among available lipoglycopeptides, dalbavancin’s pharmacokinetics and phar-macodynamics were examined for children younger than 6 years old and a single dose found to have adequate pharmacodynamic target attainment (Carrothers 2023).

Please see lines 366-375.

Round 2

Reviewer 1 Report

Thanks for revising the manuscript. I’m afraid there are a few more points to consider in the revised manuscript.

1.      Lines 33-35: The in-text citation of reference [8] is missing. If it is not used, please remove it from the references list.

2.      Line 160 (Section 3): Please replace “EOS” with “sepsis” because Section 3.3. is about late-onset sepsis “LOS”

3.      Line 470: I would suggest adding the following sentence before Table 3: The current guidelines for antibiotic use in neonatal sepsis are summarized in Table 3.

Author Response

Answers to the reviewers' comments:

We would like to thank the editor and all the reviewers for their constructive comments on our paper. We have outlined our responses to the various points raised by the reviewers sequentially and have indicated changes made to the manuscript as lines. In the revised manuscript all the changes are highlighted in red.

Comments and Suggestions for Authors

Thanks for revising the manuscript. I’m afraid there are a few more points to consider in the revised manuscript.

  1. Lines 33-35: The in-text citation of reference [8] is missing. If it is not used, please remove it from the references list.

Reply:

We thank the reviewer for raising this point.

The reference 8 has been appropriately cited in the text.

Please see lines 33-35.

  1. Line 160 (Section 3): Please replace “EOS” with “sepsis” because Section 3.3. is about late-onset sepsis “LOS”

Reply:

We thank the reviewer for raising this point.

We have revised EOS with sepsis in the section title 3.

Please see line 161.

  1. Line 470: I would suggest adding the following sentence before Table 3: The current guidelines for antibiotic use in neonatal sepsis are summarized in Table 3.

Reply:

We thank the reviewer for raising this point.

We have added the sentence ‘The current guidelines for antibiotic use in neonatal sepsis are summarized in Table 3before table 3.

Please see lines 470-471.

Reviewer 3 Report

Thanks for this revision. Once again, I really liked reading this paper, it is very well structured. Yet, in terms of your statement re. meningitis treatment; regimens usually consist of either ampicillin/amoxicillin (covers Listeria) AND a cephalosporine. Aminoglycosides hamper CSF penetration. It is true though that some guidelines and experts advocate for a short course of gent during the treatment for GBS meningitis, but it is definitely not routinely recommended to give ampicillin, cephalosporins AND aminoglycosides for every neonate (see AAP guideline, CPS statement, NICE guidance). Please revise this.

Another minor suggestion:

- Line 14: would add "abx are the most FREQUENTLY prescribed drugs"

Author Response

Answers to the reviewers' comments:

We would like to thank the editor and all the reviewers for their constructive comments on our paper. We have outlined our responses to the various points raised by the reviewers sequentially and have indicated changes made to the manuscript as lines. In the revised manuscript all the changes are highlighted in red.

Comments and Suggestions for Authors

Thanks for this revision. Once again, I really liked reading this paper, it is very well structured. Yet, in terms of your statement re. meningitis treatment; regimens usually consist of either ampicillin/amoxicillin (covers Listeria) AND a cephalosporine. Aminoglycosides hamper CSF penetration. It is true though that some guidelines and experts advocate for a short course of gent during the treatment for GBS meningitis, but it is definitely not routinely recommended to give ampicillin, cephalosporins AND aminoglycosides for every neonate (see AAP guideline, CPS statement, NICE guidance). Please revise this.

Reply:

We thank the reviewer for raising this important this point.

We have revised the part of the manuscript regarding the treatment of neonatal meningitis to: ‘The combination of ampicillin and gentamicin is the most appropriate treatment for the most common organisms, GBS and Escherichia coli [58, 60, 69], while if meningitis is suspected, an expanded-spectrum cephalosporin such as cefotaxime should substitute the aminoglycoside [70, 71].’

Please see lines 243-247, and Table 3.

Another minor suggestion:

- Line 14: would add "abx are the most FREQUENTLY prescribed drugs"

Reply:

We thank the reviewer for this point.

We have revised the phrase and added ‘frequently’.

Please see line 14.

Round 3

Reviewer 3 Report

Thanks for this revision.

Suggest acceptance in present form.